# Active Bias: Training More Accurate Neural Networks by Emphasizing High Variance Samples

**Haw-Shiuan Chang, Erik Learned-Miller, Andrew McCallum**
University of Massachusetts, Amherst
140 Governors Dr., Amherst, MA 01003
{hschang,elm,mccallum}@cs.umass.edu

## Abstract

*Self-paced learning* and *hard example mining* re-weight training instances to improve learning accuracy. This paper presents two improved alternatives based on lightweight estimates of sample uncertainty in stochastic gradient descent (SGD): the variance in predicted probability of the correct class across iterations of mini-batch SGD, and the proximity of the correct class probability to the decision threshold. Extensive experimental results on six datasets show that our methods reliably improve accuracy in various network architectures, including additional gains on top of other popular training techniques, such as residual learning, momentum, ADAM, batch normalization, dropout, and distillation.

## 1 Introduction

Learning easier material before harder material is often beneficial to human learning. Inspired by this observation, *curriculum learning* [5] has shown that learning from easier instances first can also improve neural network training. When it is not known *a priori* which samples are easy, examples with lower loss on the current model can be inferred to be easier and can be used in early training. This strategy has been referred to as *self-paced learning* [25]. By decreasing the weight of difficult examples in the loss function, the model may become more robust to outliers [33], and this method has proven useful in several applications, especially with noisy labels [36].

Nevertheless, selecting easier examples for training often slows down the training process because easier samples usually contribute smaller gradients, and the current model has already learned how to make correct predictions on these samples. On the other hand, and somewhat ironically, the opposite strategy (i.e., sampling harder instances more often) has been shown to accelerate (mini-batch) stochastic gradient descent (SGD) in some cases, where the difficulty of an example can be defined by its loss [18, 29, 44] or be proportional to the magnitude of its gradient [51, 1, 12, 13]. This strategy is sometimes referred to as *hard example mining* [44].

In the literature, we can see that these two opposing strategies work well in different situations. Preferring easier examples may be effective when either machines or humans try to solve a challenging task containing more label noise or outliers. On the other hand, focusing on harder samples may accelerate and stabilize SGD in cleaner data by minimizing the variance of gradients [1, 12]. However, we often do not know how noisy our training dataset is. Motivated by this practical need, this paper explores new methods of re-weighting training examples that are effective in both scenarios.

Intuitively, if a model has already predicted some examples correctly with high confidence, those samples may be too easy to contain useful information for improving that model further. Similarly, if some examples are always predicted incorrectly over many iterations of training, these examples may just be too difficult/noisy and may degrade the model. This suggests that we should somehow prefer uncertain samples that are predicted incorrectly sometimes during training and correctly at

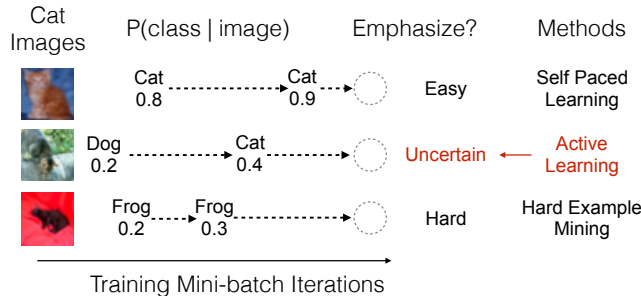

Figure 1: The proposed methods emphasize uncertain samples based on previous prediction history.

other times, as illustrated in Figure 1. This preference is consistent with common variance reduction strategies in *active learning* [43].

Previous studies suggest that finding informative unlabeled samples to label is related to selecting already-labeled samples to optimize the model parameters [14]. As reported in the previous studies [42, 6], models can sometimes achieve lower generalization error after being trained with only a subset of actively selected training data. In other words, focusing on informative samples can be beneficial even when all labels are available.

We propose two lightweight methods that actively emphasize uncertain samples to improve mini-batch SGD for classification. One method measures the variance of prediction probabilities, while the other one estimates the closeness between the prediction probabilities and the decision threshold. For logistic regression, both methods can be proven to reduce the uncertainty in the model parameters under reasonable approximations.

We present extensive experiments on CIFAR 10, CIFAR 100, MNIST (image classification), Question Type (sentence classification), CoNLL 2003, and OntoNote 5.0 (Named entity Recognition), as well as on different architectures, including multiple class logistic regression, fully-connected networks, convolutional neural networks (CNNs) [26], and residual networks [16]. The results show that active bias makes neural networks more robust without prior knowledge of noise, and reduces the generalization error by 1% –18% even on training sets having few (if any) annotation errors.

## 2   Related work

As (deep) neural networks become more widespread, many methods have recently been proposed to improve SGD training. When using (mini-batch) SGD, the randomness of the gradient sometimes slows down the optimization, so one common approach is to use the gradient computed in previous iterations to stabilize the process. Examples include momentum [38], stochastic variance reduced gradient (SVRG) [21], and proximal stochastic variance reduced gradient (Prox-SVRG) [49]. Other work proposes variants of semi-stochastic algorithms to approximate the exact gradient direction and reduce the gradient variance [47, 34]. More recently, supervised optimization methods like *learning by learning* [3] also show great potential in this problem.

In addition to the high variance of the gradient, another issue with SGD is the difficulty of tuning the learning rate. Like Quasi-Newton methods, several methods adaptively adjust learning rates based on local curvature [2, 40], while ADAGRAD [11] applies different learning rates to different dimensions. ADAM [23] combines several of these techniques and is widely used in practice.

More recently, some studies accelerate SGD by weighting each class differently [13] or weighting each sample differently as we do [18, 51, 29, 12, 1, 44], and their experiments suggest that the methods are often compatible with other techniques such as Prox-SVRG, ADAGRAD, or ADAM [29, 13]. Notice that Gao et al. [12] discuss the idea of selecting uncertain examples for SGD based on active learning, but their proposed methods choose each sample according to the magnitude of its gradient as in ISSGD [1], which actually prefers more difficult examples.

The aforementioned methods focus on accelerating the optimization of a fixed loss function given a fixed model. Many of these methods adopt importance sampling. That is, if the method prefers to

select harder examples, the learning rate corresponding to those examples will be lower. This makes gradient estimation unbiased [18, 51, 1, 12, 13], which guarantees convergence [51, 13].

On the other hand, to make models more robust to outliers, some approaches inject bias into the loss function in order to emphasize easier examples [37, 48, 27, 35]. Some variants of the strategy gradually increase the loss of hard examples [32], as in self-paced learning [25]. To alleviate the local minimum problem during training, other techniques that smooth the loss function have been proposed recently [8, 15]. Nevertheless, to our knowledge, it remains an unsolved challenge to balance the easy and difficult training examples to facilitate training while remaining robust to outliers.

## 3  Methods

In this section, we first discuss the baseline methods against which we shall compare and introduce some notations which we are going to use later on. We then present our two active bias methods based on prediction variance and closeness to the decision threshold.

### 3.1  Baselines

Due to its simplicity and generally good performance, the most widely used version of SGD samples each training instance uniformly. This basic strategy has two variants. The first samples with replacement. Let $\mathcal{D} = (\mathbf{x_i}, y_i)_i$ indicate the training dataset. The probability of selecting each sample is equal (i.e., $P_s(i|\mathcal{D}) = \frac{1}{|\mathcal{D}|}$), so we call it *SGD Uniform* (SGD-Uni). The second samples without replacement. Let $S_e$ be the set of samples we have already used in the current epoch. Then, the sampling probability $P_s(i|S_e, \mathcal{D})$ would become $(\frac{1}{|\mathcal{D}|-|S_e|})\mathbf{1}_{i \notin S_e}$, where $\mathbf{1}$ is an indicator function. This version scans through all of the examples in each epoch, so we call it *SGD-Scan*.

We propose a simple baseline which selects harder examples with higher probability, as done by Loshchilov and Hutter [29]. Specifically, we let $P_s(i|H, S_e, \mathcal{D}) \propto 1 - \bar{p}_{H_i^{t-1}}(y_i|\mathbf{x_i}) + \epsilon_D$, where $H_i^{t-1}$ is the history of prediction probability which stores all $p(y_i|\mathbf{x_i})$ when $\mathbf{x_i}$ is selected to train the network before the current iteration $t$, $H = \bigcup_i H_i^{t-1}$, $\bar{p}_{H_i^{t-1}}(y_i|\mathbf{x_i})$ is the average probability of classifying sample $i$ into its correct class $y_i$ over all the stored $p(y_i|\mathbf{x_i})$ in $H_i^{t-1}$, and $\epsilon_D$ is a smoothness constant. Notice that by only considering $p(y_i|\mathbf{x_i})$ in $H_i^{t-1}$, we won't need to perform extra forward passes. We refer to this simple baseline as *SGD Sampled by Difficulty* (SGD-SD).

In practice, SGD-Scan often works better than SGD-Uni because it ensures that the model sees all of the training examples in each epoch. To emphasize difficult examples while applying SGD-Scan, we weight each sample differently in the loss function. That is, the loss function is modified as $L = \sum_i v_i \cdot loss_i(W) + \lambda R(W)$, where $W$ are the parameters in the model, $loss_i(W)$ is the prediction loss, and $\lambda R(W)$ is the regularization term of the model. The weight of the $i$th sample $v_i$ can be set as $\frac{1}{N_D}(1 - \bar{p}_{H_i^{t-1}}(y_i|\mathbf{x_i}) + \epsilon_D)$, where $N_D$ is a normalization constant making the average of $v_i$ equal to 1. We want to keep the average of the $v_i$ fixed so that we do not change the global learning rate. We denote this method *SGD Weighted by Difficulty* (SGD-WD).

Models usually cannot fit outliers well, so SGD-SD and SGD-WD would not be robust to noise. To make a model unbiased, importance sampling can be used. That is, we can let $P_s(i|H, S_e, \mathcal{D}) \propto 1 - \bar{p}_{H_i^{t-1}}(y_i|\mathbf{x_i}) + \epsilon_D$ and $v_i = N_D(1 - \bar{p}_{H_i^{t-1}}(y_i|\mathbf{x_i}) + \epsilon_D)^{-1}$, which is similar to an approach used by Hinton [18]. We refer to this as *SGD Importance-Sampled by Difficulty* (SGD-ISD).

In addition, we propose two simple baselines that emphasize easy examples, as in self-paced learning. Based on the same naming convention, *SGD Sampled by Easiness* (SGD-SE) denotes that $P_s(i|H, S_e, \mathcal{D}) \propto \bar{p}_{H_i^{t-1}}(y_i|\mathbf{x_i}) + \epsilon_E$, while *SGD Weighted by Easiness* (SGD-WE) sets $v_i = \frac{1}{N_E}(\bar{p}_{H_i^{t-1}}(y_i|\mathbf{x_i}) + \epsilon_E)$, where $N_E$ normalizes the $v_i$'s to have unit mean.

### 3.2  Prediction Variance

In the active learning setting, the prediction variance can be used to measure the uncertainty of each sample for either a regression or classification problem [41]. In order to gain more information at each SGD iteration, we choose samples with high prediction variances.

Since the prediction variances are estimated on the fly, we would like to balance exploration and exploitation. Adopting the *optimism in face of uncertainty* heuristics of bandit problems [7], we draw the next sample based on the estimated prediction variance plus its confidence interval. Specifically, for *SGD Sampled by Prediction Variance* (SGD-SPV), we let

$$P_s(i|H, S_e, \mathcal{D}) \propto \widehat{std}_i^{\text{conf}}(H) + \epsilon_V, \text{where} \quad \widehat{std}_i^{\text{conf}}(H) = \sqrt{\widehat{var}(p_{H_i^{t-1}}(y_i|\mathbf{x_i})) + \frac{\widehat{var}(p_{H_i^{t-1}}(y_i|\mathbf{x_i}))^2}{|H_i^{t-1}| - 1}}, \tag{1}$$

$\widehat{var}\left(p_{H_i^{t-1}}(y_i|\mathbf{x_i})\right)$ is the prediction variance estimated by history $H_i^{t-1}$, and $|H_i^{t-1}|$ is the number of stored prediction probabilities. Assuming $p_{H_i^{t-1}}(y_i|\mathbf{x_i})$ is normally distributed under the uncertainty of model parameters $\mathbf{w}$, the variance of prediction variance estimation can be estimated by $2 \cdot \widehat{var}\left(p_{H_i^{t-1}}(y_i|\mathbf{x_i})\right)^2 (|H_i^{t-1}| - 1)^{-1}$. As we did in the baselines, adding the smoothness constant $\epsilon_V$ prevents the low variance instances from never being selected again. Similarly, another variant of the method sets $v_i = \frac{1}{N_V}(\widehat{std}_i^{\text{conf}}(H) + \epsilon_V)$, where $N_V$ normalizes $v_i$ like other weighted methods; we call this *SGD Weighted by Prediction Variance* (SGD-WPV).

As in SGD-WD, SGD-WE or self-paced learning [4], we train an unbiased model for several burn-in epochs at the beginning so as to judge the sampling uncertainty reasonably and stably. Other implementation details will be described in the first section of the supplementary material.

Using a low learning rate, model parameters $\mathbf{w}$ would be close to a good local minimum after sufficient burn-in epochs, and thus the posterior distribution of $\mathbf{w}$ can be locally approximated by a Gaussian distribution. Furthermore, the prediction distribution $p(y_i|\mathbf{x_i}, \mathbf{w})$ is often locally smooth with respect to the model parameters $\mathbf{w}$ (i.e., small changes of model parameters only induce small changes in the prediction distribution), so a Gaussian tends to approximate the distribution of $p_{H_i^{t-1}}(y_i|\mathbf{x_i})$ well in practice.

**Example: logistic regression**

Given a Gaussian prior $Pr(W = \mathbf{w}) = \mathcal{N}(\mathbf{w}|\mathbf{0}, s_0 I)$ on the parameters, consider the probabilistic interpretation of logistic regression:

$$-\log(Pr(Y, W = \mathbf{w}|X)) = -\sum_i \log(p(y_i|\mathbf{x_i}, \mathbf{w})) - \frac{c}{s_0}||\mathbf{w}||^2, \tag{2}$$

where $p(y_i|\mathbf{x_i}, \mathbf{w}) = \frac{1}{1+exp(-y_i\mathbf{w}^T\mathbf{x_i})}$, and $y_i \in \{1, -1\}$.

Since the posterior distribution of $W$ is log-concave [39], we can use $Pr(W = \mathbf{w}|Y, X) \approx \mathcal{N}(\mathbf{w}|\mathbf{w_N}, S_N)$, where $\mathbf{w_N}$ is maximum a posteriori (MAP) estimation, and

$$S_N^{-1} = \nabla_\mathbf{w} \nabla_\mathbf{w} - \log(Pr(Y, W|X)) = \sum_i p(y_i|\mathbf{x_i})\,(1 - p(y_i|\mathbf{x_i}))\,\mathbf{x_i}\mathbf{x_i}^T + \frac{2c}{s_0}I. \tag{3}$$

Then, we further approximate $p(y_i|\mathbf{x_i}, W)$ using the first order Taylor expansion $p(y_i|\mathbf{x_i}, W) \approx p(y_i|\mathbf{x_i}, \mathbf{w}) + g_i(\mathbf{w})^T(W - \mathbf{w})$, where $g_i(\mathbf{w}) = p(y_i|\mathbf{x_i}, \mathbf{w})\,(1 - p(y_i|\mathbf{x_i}, \mathbf{w}))\,\mathbf{x_i}$. We can compute the prediction variance [41] with respect to the uncertainty of $W$

$$Var(p(y_i|\mathbf{x_i}, W)) \approx g_i(\mathbf{w})^T S_N g_i(\mathbf{w}). \tag{4}$$

These approximations tell us several things. First, $Var(p(y_i|\mathbf{x_i}, W))$ is proportional to $p(y_i|\mathbf{x_i}, \mathbf{w})^2(1 - p(y_i|\mathbf{x_i}, \mathbf{w}))^2$, so the prediction variance is larger when the sample $i$ is closer to the boundary. Second, when we have more sample points close to the boundary, the variance of the parameters $S_N$ is lower. That is, when we emphasize samples with high prediction variances, the uncertainty of parameters tends to be reduced, akin to the variance reduction strategy in active learning [30]. Third, with a Gaussian assumption on the posterior distribution $Pr(W = \mathbf{w}|Y, X)$ and the Taylor expansion, the distribution of $p(y_i|\mathbf{x_i}, W)$ in logistic regression becomes Gaussian, which justifies our previous assumption of $p_{H_i^{t-1}}(y_i|\mathbf{x_i})$ for the confidence estimation of the prediction variance. Notice that there are other methods that can measure the prediction uncertainty, such as the mutual information between labels and parameters [19], but we found that the prediction variance works better in our experiments.

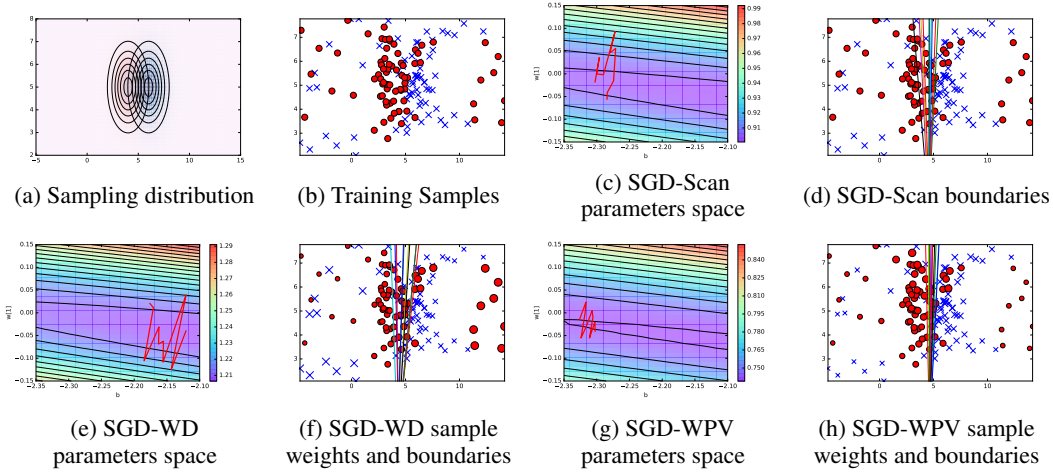

(a) Sampling distribution    (b) Training Samples    (c) SGD-Scan parameters space    (d) SGD-Scan boundaries

(e) SGD-WD parameters space    (f) SGD-WD sample weights and boundaries    (g) SGD-WPV parameters space    (h) SGD-WPV sample weights and boundaries

Figure 2: A toy example which compares different methods in a two-class logistic regression model. To visualize the optimization path for the classifier parameters (the red paths in (c), (e), and (g)) in two dimensions, we fix the weight corresponding to the x-axis to 0.5 and only show the weight for y-axis $w[1]$ and bias term $b$. The $i$th sample size in (f) and (h) is proportional to $v_i$. The toy example shows that SGD-WPV can train a more accurate model in a noisy dataset.

Figure 2 illustrates a toy example. Given the same learning rate, we can see that the normal SGD in Figure 2c and 2d will have higher uncertainty when there are many outliers, and emphasizing difficult examples in Figure 2e and 2f makes it worse. On the other hand, the samples near the boundaries would have higher prediction variances (i.e., larger circles or crosses in Figure 2h) and thus higher impact on the loss function in SGD-WPV.

After burn-in epochs, $\mathbf{w}$ becomes close to a local minimum using SGD. Then, the parameters estimated in each iteration can be viewed, approximately, as samples drawn from the posterior distribution of the parameters $Pr(W = \mathbf{w}|Y, X)$ [31]. Therefore, after running SGD long enough, $\widehat{var}\left(p_{H_i^{t-1}}(y_i|\mathbf{x_i})\right)$ can be used to approximate $Var\left(p(y_i|\mathbf{x_i}, W)\right)$. Notice that if we directly apply bias at the beginning without running burn-in epochs, incorrect examples might be emphasized, which is also known as the local minimum problem in active learning [14]. For instance, in Figure 2, if burn-in epochs are not applied and the initial $\mathbf{w}$ is a vertical line on the left, the outliers close to the initial boundary would be emphasized, which slows down the convergence speed.

In this simple example, we can also see that the gradient magnitude is proportional to the difficulty because $-\nabla_{\mathbf{w}}\log(p(y_i|\mathbf{x_i}, \mathbf{w})) = (1 - p(y_i|\mathbf{x_i}, \mathbf{w}))\,\mathbf{x_i}$. This is why we believe the SGD acceleration methods based on gradient magnitude [1, 13] can be categorized as variants of preferring difficult examples, and thus more vulnerable to outliers (like the samples on the left or right in Figure 2).

### 3.3 Threshold Closeness

Motivated by the previous analysis, we propose a simpler and more direct approach to select samples whose correct class probability is close to the decision threshold. *SGD Sampled by Threshold Closeness* (SGD-STC) makes $P_s(i|H, S_e, \mathcal{D}) \propto \bar{p}_{H_i^{t-1}}(y_i|\mathbf{x_i})\left(1 - \bar{p}_{H_i^{t-1}}(y_i|\mathbf{x_i})\right) + \epsilon_T$, where $\bar{p}_{H_i^{t-1}}(y_i|\mathbf{x_i})$ is the average probability of classifying sample $i$ into its correct class $y_i$ over all the stored $p(y_i|\mathbf{x_i})$ in $H_i^{t-1}$. When there are multiple classes, this measures the closeness of the threshold for distinguishing the correct class out of the union of the rest of the classes (i.e., one-versus-rest). The method is similar to an approximation of the optimal allocation in stratified sampling proposed by Druck and McCallum [10].

Similarly, *SGD Weighted by Threshold Closeness* (SGD-WTC) chooses the weight of $i$th sample $v_i = \frac{1}{N_T}\bar{p}_{H_i^{t-1}}(y_i|\mathbf{x_i})(1-\bar{p}_{H_i^{t-1}}(y_i|\mathbf{x_i}))+\epsilon_T$, where $N_T = \frac{1}{|\mathcal{D}|}\sum_j \bar{p}_{H_j^{t-1}}(y_j|\mathbf{x_j})(1-\bar{p}_{H_j^{t-1}}(y_j|\mathbf{x_j}))+\epsilon_T$. The weighting can be viewed as combining the SGD-WD and SGD-WE by multiplying their weights

Table 1: Model architectures. Dropouts and L2 reg (regularization) are only applied to the fully-connected (FC) layer(s).

| Dataset | # Conv layers | Filter size | Filter number | # Pooling layers | # BN layers | # FC layers | Dropout keep probs | L2 reg |
|---|---|---|---|---|---|---|---|---|
| MNIST | 2 | 5x5 | 32, 64 | 2 | 0 | 2 | 0.5 | 0.0005 |
| CIFAR 10 | 0 | N/A | N/A | 0 | 0 | 1 | 1 | 0.01 |
| CIFAR 100 | 26 or 62 | 3X3 | 16, 32, 64 | 0 | 13 or 31 | 1 | 1 | 0 |
| Question Type | 1 | (2,3,4)x1 | 64 | 1 | 0 | 1 | 0.5 | 0.01 |
| CoNLL 2003 OntoNote 5.0 | 3 | 3x1 | 100 | 0 | 0 | 1 | 0.5, 0.75 | 0.001 |
| MNIST | 0 | N/A | N/A | 0 | 0 | 2 | 1 | 0 |

Table 2: Optimization hyper-parameters and experiment settings

| Dataset | Optimizer | Batch size | Learning rate | Learning rate decay | # Epochs | # Burn-in epochs | # Trials |
|---|---|---|---|---|---|---|---|
| MNIST | Momentum | 64 | 0.01 | 0.95 | 80 | 2 | 20 |
| CIFAR 10 | SGD | 100 | 1e-6 | 0.5 (per 5 epochs) | 30 | 10 | 30 |
| CIFAR 100 | Momentum | 128 | 0.1 | 0.1 (at 80, 100, 120 epochs) | 150 | 90 or 50 | 20 |
| Question Type | ADAM | 64 | 0.001 | 1 | 250 | 50 | 100 |
| CoNLL 2003 OntoNote 5.0 | ADAM | 128 | 0.0005 | 1 | 200 | 30 | 10 |
| MNIST | SGD | 128 | 0.1 | 1 | 60 | 20 | 10 |

together. Although other uncertainty estimates such as entropy are widely used in active learning and can also be viewed as a measure of boundary closeness, we found the proposed formula works better in our experiments.

When using logistic regression, after injecting the bias $v_i$ into the loss function, approximating the prediction probability based on previous history, removing the regularization and smoothness constant (i.e., $p(y_i|\mathbf{x_i}, \mathbf{w}) \approx \bar{p}_{H_i^{t-1}}(y_i|\mathbf{x_i})$, $1/s_0 = 0$, and $\epsilon_T = 0$), we can show that

$$\sum_i Var(p(y_i|\mathbf{x_i}, W)) \approx \sum_i g_i(\mathbf{w})^T S_N g_i(\mathbf{w}) \approx N_T \cdot dim(w), \tag{5}$$

where $dim(w)$ is the dimension of parameters $w$. This will ensure that the average prediction variance drops linearly as the number of training instance increases. The derivation could be seen in the supplementary materials.

## 4 Experiments

We test our methods on six different datasets. The results show that the active bias techniques constantly outperform the standard uniform sampling (i.e., SGD-Uni and SGD-Scan) in the deep models as well as the shallow models. For each dataset, we use an existing, publicly available implementation for the problem and emphasize samples using different methods. The architectures and hyper-parameters are summarized in Table 1. All neural networks use softmax and cross-entropy loss at the last layer. The optimization and experiment setups are listed in Table 2. As shown in the second column of the table, SGD in CNNs and residual networks actually refers to momentum or ADAM instead of vanilla SGD. All experiments use mini-batch.

Like most of the widely used neural network training techniques, the proposed techniques are not applicable to every scenario. For all the datasets we tried, we found that the proposed methods are not sensitive to the hyper-parameter setup except when applying a very complicated model to a relatively smaller dataset. If a complicated model achieves 100% training accuracy within a few epochs, the most uncertain examples would often be outliers, biasing the model towards overfitting.

To avoid this scenario, we modify the default hyper-parameters setup in the implementation of the text classifiers in Section 4.3 and Section 4.4 to achieve similar performance using simplified models. For all other models and datasets, we use the default hyper-parameters of the existing implementations, which should favor the SGD-Uni or SGD-Scan methods, since the default hyper-parameters are

Table 3: The average of the best testing error rates for different sampling methods and datasets (%). The confidence intervals are standard errors. LR means logistic regression.

| Datasets | Model | SGD-Uni | SGD-SD | SGD-ISD | SGD-SE | SGD-SPV | SGD-STC |
|---|---|---|---|---|---|---|---|
| MNIST | CNN | 0.55±0.01 | 0.52±0.01 | 0.57±0.01 | 0.54±0.01 | **0.51** ±0.01 | **0.51**±0.01 |
| Noisy MNIST | CNN | 0.83±0.01 | 1.00±0.01 | 0.84±0.01 | 0.69 ±0.01 | 0.64±0.01 | **0.63**±0.01 |
| CIFAR 10 | LR | 62.49±0.06 | 63.14±0.06 | 62.48±0.07 | 60.87±0.06 | **60.66**±0.06 | 61.00±0.06 |
| QT | CNN | 17.70±0.07 | 17.61±0.07 | 17.66±0.08 | 17.92±0.08 | **17.49**±0.08 | 17.55±0.08 |

Table 4: The average of the best testing error rates and their standard errors for different weighting methods (%). For CoNLL 2003 and OntoNote 5.0, the values are 1-(F1 score). CNN, LR, RN 27, RN 63 and FC mean convolutional neural network, logistic regression, residual networks with 27 layers, residual network with 63 layers, and fully-connected network, respectively.

| Datasets | Model | SGD-Scan | SGD-WD | SGD-WE | SGD-WPV | SGD-WTC |
|---|---|---|---|---|---|---|
| MNIST | CNN | 0.54±0.01 | **0.48**±0.01 | 0.56±0.01 | **0.48**±0.01 | **0.48**±0.01 |
| Noisy MNIST | CNN | 0.81±0.01 | 0.92±0.01 | 0.72±0.01 | **0.61**±0.02 | 0.63±0.01 |
| CIFAR 10 | LR | 62.48±0.06 | 63.10±0.06 | 60.88±0.06 | **60.61**±0.06 | 61.02±0.06 |
| CIFAR 100 | RN 27 | 34.04±0.06 | 34.55±0.06 | 33.65±0.07 | 33.69±0.07 | **33.64**±0.07 |
| CIFAR 100 | RN 63 | 30.70±0.06 | 31.57±0.09 | **29.92**±0.09 | 30.02±0.08 | 30.16±0.09 |
| QT | CNN | 17.79±0.08 | 17.70±0.08 | 17.87±0.08 | **17.57**±0.07 | 17.61±0.08 |
| CoNLL 2003 | CNN | 11.62±0.04 | 11.50±0.05 | 11.73±0.04 | 11.24±0.06 | **11.18**±0.03 |
| OntoNote 5.0 | CNN | 17.80±0.05 | 17.65±0.06 | 18.40±0.05 | 17.82±0.03 | **17.51**±0.05 |
| MNIST | FC | 2.85±0.03 | **2.17**±0.01 | 3.08±0.03 | 2.68±0.02 | 2.34±0.03 |
| MNIST (distill) | FC | 2.27±0.01 | 2.13±0.02 | 2.35±0.01 | 2.18±0.02 | **2.07**±0.02 |

optimized for these cases. To show the reliability of the proposed methods, we do not optimize the hyper-parameters for the proposed methods or baselines.

Due to the randomness in all the SGD variants, we repeat experiments and list the number of trials in Table 2. At the beginning of each trial, network weights are trained with uniform sampling SGD until validation performance starts to saturate. After these burn-in epochs, we apply different sampling/weighting methods and compare performance. The number of burn-in epochs is determined by cross-validation, and the number of epochs in each trial is set large enough to let the testing error of most methods converge. In Tables 3 and 4, we evaluate the testing performance of each method after each epoch and report the best testing performance among epochs within each trial.

As previously discussed, there are various versions preferring easy or difficult examples. Some of them require extra time to collect necessary statistics such as the gradient magnitude of each sample [12, 1], change the network architecture [15, 44], or involve an annealing schedule like self-paced learning [25, 32]. We tried self-paced learning on CIFAR 10 but found that performance usually remains the same and is sometimes sensitive to the hyper-parameters of the annealing schedule. This finding is consistent with the results from [4]. To simplify the comparison, we focus on testing the effects of steady bias based on sample difficulty (e.g., compare with SGD-SE and SGD-SD) and do not gradually change the preference during the training like self-paced learning.

It is not always easy to change the sampling procedure because of the model or implementation constraints. For example, in sequence labeling tasks (CoNLL 2003 and OntoNote 5.0), the words in the same sentence need to be trained together. Thus, we only compare methods which modify the loss function (SGD-W*) with SGD-Scan for some models. For the other experiments, re-weighting examples (SGD-W*) generally gives us better performance than changing the sampling distribution (SGD-S*). It might be because we can better estimate the statistics of each sample.

## 4.1 MNIST

We apply our method to a CNN [26] for MNIST[1] using one of the Tensorflow tutorials.[2] The dataset has high testing accuracy, so most of the examples are too easy for the model after a few epochs. Selecting more difficult instances can accelerate learning or improve testing accuracy [18, 29, 13]. The results from SGD-SD and SGD-WD confirm this finding while selecting uncertain examples can give us a similar or larger boost. Furthermore, we test the robustness of our methods by randomly

reassigning the labels of 10% of the images, and the results indicate that the SGD-WPV improves the performance of SGD-Scan even more while SGD-SD overfits the data seriously.

## 4.2 CIFAR 10 and CIFAR 100

We test a simple multi-class logistic regression[3] on CIFAR 10 [24].[4] Images are down-sampled significantly to $32 \times 32 \times 3$, so many examples are difficult, even for humans. SGD-SPV and SGD-SE perform significantly better than SGD-Uni here, consistent with the idea that avoiding difficult examples increases robustness to outliers.

For CIFAR 100 [24], we demonstrate that the proposed approaches can also work in very deep residual networks [16].[5] To show the method is not sensitive to the network depth and the number of burn-in epochs, we present results from the network with 27 layers and 90 burn-in epochs as well as the network with 63 layers and 50 burn-in epochs. Without changing architectures, emphasizing uncertain or easy examples gains around 0.5% in both settings, which is significant considering the fact that the much deeper network shows only 3% improvement here.

When training a neural network, gradually reducing the learning rate (i.e., the magnitude of gradients) usually improves performance. When difficult examples are sampled less, the magnitude of gradients would be reduced. Thus, some of the improvement of SGD-SPV and SGD-SE might come from using a lower effective learning rate. Nevertheless, since we apply the aggressive learning rate decay in the experiments of CIFAR 10 and CIFAR 100, we know that the improvements from SGD-SPV and SGD-SE cannot be entirely explained by its lower effective learning rate.

## 4.3 Question Type

To investigate whether our methods are effective for smaller text datasets, we apply them to a sentence classification dataset (i.e. Question Type (QT) [28]), which contains 1000 training examples and 500 testing examples.[6] We use the CNN architecture proposed by Kim [22].[7] Like many other NLP tasks, the dataset is relatively small and this CNN classifier does not inject noise to inputs like the implementation of residual networks in CIFAR 100, so this complicated model reaches 100% training accuracy within a few epochs.

To address this, we reduced the model complexity by (i) decreasing the number of filters from 128 to 64, (ii) decreasing convolutional filter widths from 3,4,5 to 2,3,4, (iii) adding L2 regularization with scale 0.01, (iv) performing PCA to reduce the dimension of pre-trained word embedding from 300 to 50 and fixing the word embedding during training. Then, the proposed active bias methods perform better than other baselines in this smaller model.

## 4.4 Sequence Tagging Tasks

We also test our methods on Named Entity Recognition (NER) in CoNLL 2003 [46] and OntoNote 5.0 [20] datasets using the CNN from Strubell et al. [45].[8] Similar to Question Type, the model is too complex for our approaches. So we (i) only use 3 layers instead of 4 layers, (ii) reduce the number of filters from 300 to 100, (iii) add 0.001 L2 regularization, (iv) make the 50 dimension word embedding from Collobert et al. [9] non-trainable. The micro F1 of this smaller model only drops around 1%-2% from the original big model. Table 4 shows that our methods achieve the lowest error rate (1-F1) in both benchmarks.

## 4.5 Distillation

Although state-of-the-art neural networks in many applications memorize examples easily [50], much simpler models can usually achieve similar performance like those in the previous two experiments. In practice, such models are often preferable due to their low computation and memory requirements.

We have shown that the proposed method can improve these smaller models as distillation did [17], so it is natural to check whether our methods can work well with distillation. We use an implementation[9] that distills a shallow CNN with 3 convolution layers to a 2 layer fully-connected network in MNIST. The teacher network can achieve 0.8% testing error, and the temperature of softmax is set as 1.

Our approaches and baselines simply apply the sample dependent weights $v_i$ to the final loss function (i.e., cross-entropy of the true labels plus cross-entropy of the prediction probability from the teacher network). In MNIST, SGD-WTC and SGD-WD can achieve similar or better improvements compared with adding distillation into SGD-Scan. Furthermore, the best performance comes from the distillation plus SGD-WTC, which shows that active bias is compatible with distillation in this dataset.

## 5 Conclusion

Deep learning researchers often gain accuracy by employing training techniques such as momentum, dropout, batch normalization, and distillation. This paper presents a new compatible sibling to these methods, which we recommend for wide use. Our relatively simple and computationally lightweight techniques emphasize the uncertain examples (i.e., SGD-*PV and SGD-*TC).

The experiments confirm that the proper bias can be beneficial to generalization performance. When the task is relatively easy (both training and testing accuracy are high), preferring more difficult examples works well. On the contrary, when the dataset is challenging or noisy (both training and testing accuracy are low), emphasizing easier samples often lead to a better performance. In both cases, the active bias techniques consistently lead to more accurate and robust neural networks as long as the classifier does not memorize all the training samples easily (i.e., training accuracy is high but testing accuracy is low).

## Acknowledgements

This material is based on research sponsored by National Science Foundation under Grant No. 1514053 and by DARPA under agreement number FA8750-1 3-2-0020 and HRO011-15-2-0036. The U.S. Government is authorized to reproduce and distribute reprints for Governmental purposes notwithstanding any copyright notation thereon. The views and conclusions contained herein are those of the authors and should not be interpreted as necessarily representing the official policies or endorsements, either expressed or implied, of DARPA or the U.S. Government.

## Footnotes

[1] http://yann.lecun.com/exdb/mnist/

[2] https://github.com/tensorflow/models/blob/master/tutorials/image/mnist

[3] https://cs231n.github.io/assignments2016/assignment2/

[4] https://www.cs.toronto.edu/~kriz/cifar.html

[5] https://github.com/tensorflow/models/tree/master/resnet

[6] http://cogcomp.org/Data/QA/QC/

[7] https://github.com/dennybritz/cnn-text-classification-tf

[8] https://github.com/iesl/dilated-cnn-ner

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
