[Supplementary Material]

# Supplementary material for
# Active Bias: Training a More Accurate Neural Network by Emphasizing High Variance Samples

**Haw-Shiuan Chang, Erik Learned-Miller, Andrew McCallum**
University of Massachusetts, Amherst
140 Governors Dr., Amherst, MA 01003
{hschang,elm,mccallum}@cs.umass.edu

## A   Implementation details

The general framework of the methods can be seen in Algorithm 1. In each aforementioned method, if $P_s(i|H, S_e, \mathcal{D})$ is not specified, we use SGD-Scan (without replacement uniform sampling). If $v_i$ is not specified, it means $v_i = 1$ for all sample $i$. The $\epsilon$ in each method is set as the average of current estimation. For example, $\epsilon_D$ for SDG Sampled by Difficulty is set as $\frac{1}{|\mathcal{D}|} \sum_j 1 - \bar{p}_{H_j^{t-1}}(y_j|\mathbf{x_j})$.

When estimating sample related statistics like prediction variance, we found that excluding the prediction history near the beginning transient state improves performance. In our implementation, we use a simple outlier removal by computing the deviation between the prediction probability and its average at iteration $t$ (i.e., $d_t = \left| p(y_i|\mathbf{x_i}) - \bar{p}_{H_i^{t-1}}(y_i|\mathbf{x_i}) \right|$), and excludes all prediction probability $p_t(y_i|\mathbf{x_i})$ at current iteration when $d_t > 2 \cdot median_{t'}(d_{t'})$. We apply the same method when estimating difficulty, easiness, prediction variance or threshold closeness.

By only using the prediction results from previous iterations, implementing the methods is easy and the overhead of the method is very small because we do not need any extra forward or backward passes in the neural network. Due to the outlier removal process, the average overhead for each sample at each epoch is $O(E)$, where $E$ is the number of total epochs.

When we have a very large number of samples and epochs, we can modify outlier removal by only considering the prediction probability in the latest few epochs. Then, the overhead is constant. In Section 4.4, performing outlier removal in the prediction history of each word is time-consuming, so we determine the uncertainty only based on the latest 5 epochs.

---
**Algorithm 1** SGD Training with Sample Emphasis
---
**Input:** Training data $\mathcal{D}$, Batch size $|B|$, Number of class $|C|$, # epochs $E$, # burn-in epochs $e_b$
**Output:** NN
  Initialize all weights $W$ in NN
  $H_i \leftarrow \{\frac{1}{|C|}\}$ for all training sample $i$
  $v_i \leftarrow 1$ for all training sample $i$
  $t \leftarrow 1$
  **for** epoch $e \leftarrow 1...E$ **do**
    $S_e \leftarrow \emptyset$
    **for** each iteration **do**
      **if** $e > e_b$ **then**
        Sample $B$ according to $P_s(i|H, S_e, \mathcal{D})$
      **else**
        Sample $B$ uniformly from $\mathcal{D}$
      Weight sample $i$ by $v_i$ for all $i$ in $B$
      Update parameters $W$ in NN
      **for** $i$ in $B$ **do**
        $H_i \leftarrow H_i \cup \{p_t(y_i|\mathbf{x_i})\}$
        $S_e \leftarrow S_e \cup \{i\}$
        Update $P_s(i|H, S_e, \mathcal{D})$ and $v_i$.
      $t \leftarrow t + 1$
---

# B  Experiment details

Summaries of dataset properties can be seen in Table 5.

In Figure 3 and Figure 4, we present the convergence curves of MNIST without noise for the experiment in Section 4.1. By comparing the error rates, we can see that changing the sampling distribution accelerates the training more, but changing the loss function can give us better results at the end.

Figure 3: MNIST error rate (%)

Figure 4: MNIST error rate (%)

In the paper, we only provide the best testing performance within each trial. To further understand the characteristics of each method, we report the average testing performance of the last 10 epochs in Table 6 and Table 7. The results in the tables roughly follow the same trends in Table 3 and Table 4. In addition, the training errors are presented in Table 8 and Table 9. We can see that emphasizing difficult examples indeed usually increases the training accuracy, but it does not necessarily imply the improvements in the testing error.

Table 5: Dataset Statistics. The average sentence length $L$ in Question Type, CoNLL 2003, and OntoNote 5.0 datasets are 11, 14, and 18, respectively. CoNLL 2003 and OntoNote 5.0 are sequence tagging task, so each word is an instance with a label.

| Dataset | # Class | Instance | Input dimensions | # Training | # Testing |
|---|---|---|---|---|---|
| MNIST | 10 | Image | 28x28 | 60,000 | 10,000 |
| CIFAR 10 | 10 | Image | 32x32x3 | 50,000 | 10,000 |
| CIFAR 100 | 100 | Image | 32x32x3 | 50,000 | 10,000 |
| Question Type | 6 | Sentence | $50 \times L$ | 1000 | 500 |
| CoNLL 2003 | 17 | Word | $50 \times L$ | 204,567 | 46,666 |
| OntoNote 5.0 | 74 | Word | $50 \times L$ | 1,088,503 | 152,728 |

Table 6: Testing error rates (last 10 epochs) of sampling methods (%). Notice that we drop the whole rows of standard errors in the table when they are all below 0.01%.

| Datasets | Model | SGD-Uni | SGD-SD | SGD-ISD | SGD-SE | SGD-SPV | SGD-STC |
|---|---|---|---|---|---|---|---|
| MNIST | CNN | 0.59 | 0.56 | 0.60 | 0.58 | **0.55** | **0.55** |
| Noisy MNIST | CNN | 1.18±0.00 | 1.52±0.01 | 1.26±0.00 | **0.76**±0.00 | 0.92±0.00 | 0.85±0.00 |
| CIFAR 10 | LR | 62.66±0.01 | 63.35±0.01 | 62.64±0.01 | 61.01±0.01 | **60.80**±0.01 | 61.16±0.01 |
| QT | CNN | 18.56±0.01 | 18.48±0.01 | 18.51±0.01 | 18.79±0.01 | **18.33**±0.01 | 18.44±0.01 |

Table 7: Testing error rates (last 10 epochs) of sampling methods (%). For CoNLL 2003 and OntoNote 5.0, the values are 1-(F1 score). When all standard errors in a row are smaller than 0.01, we skip them in the table.

| Datasets | Model | SGD-Scan | SGD-WD | SGD-WE | SGD-WPV | SGD-WTC |
|---|---|---|---|---|---|---|
| MNIST | CNN | 0.58 | **0.51** | 0.59 | 0.53 | 0.52 |
| Noisy MNIST | CNN | 1.15±0.00 | 1.59±0.01 | **0.80**±0.00 | 0.84±0.00 | 0.85±0.00 |
| CIFAR 10 | LR | 62.61±0.01 | 63.29±0.01 | 60.99±0.01 | **60.73**±0.01 | 61.13±0.01 |
| CIFAR 100 | RN 27 | 34.21±0.01 | 34.75±0.01 | 33.82±0.02 | 33.90±0.02 | **33.81**±0.02 |
| CIFAR 100 | RN 64 | 31.06±0.01 | 32.11±0.02 | **30.17**±0.02 | 30.33±0.02 | 30.51±0.02 |
| QT | CNN | 18.59±0.01 | 18.52±0.01 | 18.68±0.01 | **18.39**±0.01 | 18.48±0.01 |
| CoNLL 2003 | CNN | 11.96±0.02 | 11.85±0.02 | 12.04±0.02 | 11.65±0.02 | **11.60**±0.02 |
| OntoNote 5.0 | CNN | 18.11±0.02 | 18.03±0.03 | 18.70±0.02 | 18.08±0.02 | **17.84**±0.02 |
| MNIST | FC | 2.91 | **2.26** | 3.15 | 2.78 | 2.41 |
| MNIST (distill) | FC | 2.33 | 2.21 | 2.41 | 2.24 | **2.14** |

## C   Proof sketch of Equation (5)

In Equation (3) and (4), by assuming

$$p(y_i|\mathbf{x_i}, W) \approx p(y_i|\mathbf{x_i}, \mathbf{w}) + g_i(\mathbf{w})^T(W - \mathbf{w}), \tag{7}$$

and

$$Pr(W = \mathbf{w}|Y, X) \approx \mathcal{N}(\mathbf{w}|\mathbf{w_N}, S_N) \tag{8}$$

, we know that

$$Var(p(y_i|\mathbf{x_i}, W)) \approx g_i(\mathbf{w})^T S_N g_i(\mathbf{w}). \tag{9}$$

We apply the

$$v_i = \frac{1}{N_T} \bar{p}_{H_i^{t-1}}(y_i|\mathbf{x_i})(1 - \bar{p}_{H_i^{t-1}}(y_i|\mathbf{x_i})) + \epsilon_T \tag{10}$$

to the loss function, so

$$L = -\sum_i v_i \log(p(y_i|\mathbf{x_i}, \mathbf{w})) - \frac{c}{s_0}||\mathbf{w}||^2. \tag{11}$$

Then, $S_N^{-1} = \sum_i v_i p(y_i|\mathbf{x_i}) (1 - p(y_i|\mathbf{x_i})) \mathbf{x_i} \mathbf{x_i}^T + \frac{2c}{s_0} I$.

Table 8: Training error rates (Best) of sampling methods (%)

| Datasets | Model | SGD-Uni | SGD-SD | SGD-ISD | SGD-SE | SGD-SPV | SGD-STC |
|---|---|---|---|---|---|---|---|
| MNIST | CNN | 0.01 | **0.00** | 0.01 | 0.05 | **0.00** | **0.00** |
| Noisy MNIST | CNN | 5.54±0.09 | **0.01**±0.00 | 2.88±0.09 | 9.08±0.01 | 7.60±0.06 | 7.83±0.04 |
| CIFAR 10 | LR | 59.88±0.02 | 60.50±0.02 | 59.88±0.02 | 58.49±0.02 | **58.26**±0.02 | 58.42±0.02 |
| QT | CNN | **0.00** | **0.00** | **0.00** | 0.04±0.01 | **0.00** | **0.00** |

Table 9: Training error rates (Best) of sampling methods (%). For CIFAR 100, the training errors are computed on the randomly cropped and flipped images.

| Datasets | Model | SGD-Scan | SGD-WD | SGD-WE | SGD-WPV | SGD-WTC |
|---|---|---|---|---|---|---|
| MNIST | CNN | 0.01 | **0.00** | 0.04 | 0.01 | 0.01 |
| Noisy MNIST | CNN | 6.21±0.15 | **0.29**±0.02 | 9.01±0.01 | 7.93±0.05 | 8.02±0.04 |
| CIFAR 10 | LR | 59.87±0.02 | 60.48±0.02 | 58.45±0.02 | **58.23**±0.02 | 58.40±0.02 |
| CIFAR 100 | RN 27 | 18.72±0.04 | **18.44**±0.04 | 19.43±0.04 | 18.86±0.04 | 18.76±0.04 |
| CIFAR 100 | RN 64 | 6.06±0.03 | **5.42**±0.04 | 8.15±0.03 | 8.41±0.03 | 7.85±0.02 |
| QT | CNN | **0.00** | **0.00** | 0.04±0.01 | **0.00** | **0.00** |
| CoNLL 2003 | CNN | 2.55±0.03 | **1.64**±0.02 | 4.00±0.03 | 2.14±0.01 | 1.86±0.02 |
| OntoNote 5.0 | CNN | 13.90±0.03 | 13.16±0.05 | 15.21±0.03 | 13.29±0.03 | **12.61**±0.03 |
| MNIST | FC | 1.84±0.01 | **0.07**±0.00 | 2.21±0.02 | 1.60±0.01 | 0.79±0.01 |
| MNIST (distill) | FC | 0.73±0.01 | **0.01**±0.00 | 0.96±0.01 | 0.58±0.01 | 0.13±0.00 |

When $p(y_i|\mathbf{x_i}, \mathbf{w}) \approx \bar{p}_{H_i^{t-1}}(y_i|\mathbf{x_i})$, $1/s_0 = 0$, and $\epsilon_T = 0$,

$$Tr(g_i(\mathbf{w})^T S_N g_i(\mathbf{w})) = Tr(g_i(\mathbf{w})g_i(\mathbf{w})^T S_N)$$

$$\approx Tr\left( \left( p(y_i|\mathbf{x_i}, \mathbf{w})^2 \left(1 - p(y_i|\mathbf{x_i}, \mathbf{w})\right)^2 \mathbf{x_i}\mathbf{x_i}^T \right) \left( \frac{1}{N_T} \sum_j p(y_j|\mathbf{x_j}, \mathbf{w})^2 \left(1 - p(y_j|\mathbf{x_j}, \mathbf{w})\right)^2 \mathbf{x_j}\mathbf{x_j}^T \right)^{-1} \right) \tag{12}$$

Finally,

$$\sum_i g_i(\mathbf{w})^T S_N g_i(\mathbf{w}) = \sum_i Tr(g_i(\mathbf{w})^T S_N g_i(\mathbf{w}))$$

$$\approx \sum_i Tr\left( p(y_i|\mathbf{x_i}, \mathbf{w})^2 \left(1 - p(y_i|\mathbf{x_i}, \mathbf{w})\right)^2 \mathbf{x_i}\mathbf{x_i}^T \left( \frac{1}{N_T} \sum_j p(y_j|\mathbf{x_j}, \mathbf{w})^2 \left(1 - p(y_j|\mathbf{x_j}, \mathbf{w})\right)^2 \mathbf{x_j}\mathbf{x_j}^T \right)^{-1} \right)$$

$$= N_T Tr(I) = N_T \cdot dim(w). \tag{13}$$