[Reviews · NeurIPS 2017]

Reviewer 1



This paper proposes a couple of alternatives to self paced learning and hard example mining: weighting or sampling data points based on a) the variance in the prediction probabilities across iterations or b) closeness of the prediction probability to the decision threshold. The paper was well written and easy to read. I like the fact that the authors test their methods on a variety of datasets and models. It seems like different methods win in different experiments though. Is there some intuition on how to choose which method to use for a given task? Have you tried or do you have any intuitions about how your methods compare against Bayesian methods for computing predictive uncertainty? E.g. Dropout at test time as proposed by Gal and Ghahramani can be used (although in some limited models) to compute the variance of predictions and could be used as an alternate to variance computed from history. Overall, this is a nice paper and the results are interesting, but not groundbreakingly so.

Reviewer 2



The paper proposes a novel way of sampling (or weighing) data-points during training of neural networks. The idea is, that one would like to sample data-point more often which could be potentially classified well but are hard to learn (in contrast to outliers or wrongly labeled ones). To `find' them the authors propose two (four if split into sampling and weighing) schemes: The first one (SGD-*PV) proposes to weigh data-points according to the variance of the predictive probability of the true label plus its confidence interval under the assumption that the prediction probability is Gaussian distributed. The second one (SGD-*TC), as far as I understand, encodes if the probability of choosing the correct label given past prediction probabilities is close to the decision threshold. The statistics needed (means and variances of p) can be computed on-the-fly during a burn-in phase of the optimizer; they can be obtained from a forward pass of the network which is computed anyways. The authors test their methods on various architectures and dataset and compare to a range of baselines. The paper motivates the problem well and the introduction is clearly written (Figure 1 is very helpful, too). The technical sections 3.2 (up to the logReg example) and 3.3 are very dense; moving table 1 (and or 2) to the appendix and elaborating a bit more (especially on 3.3) would be very helpful. Major Questions: - I do not understand how SGD-*PV prefers uncertain example over hard ones, i.e. why should hard samples have lower predictive variance than uncertain ones? Hard ones might be labeled incorrectly, but the probability might still fluctuate. - In 3.3 P(i|H, S_e, D) is largest around \bar{p}=0.5. Does that mean the method implicitly assumes 2 output classes only? If not, I would like to know how P encodes closeness to a threshold for multiple classes. - In 3.2 the authors assume that p is Gaussian distributed. Is this meant across occurrences in the mini-batch? I would also like the authors to comment a bit more on this assumption, also since clearly p is not Gaussian distributed (it ranges between 0 and 1). Would it be possible to impose this assumption on log(p) instead to get a better fit? - how many samples do you typically collect for p? as many as epochs? is this the reason you need a burn in time? or is there another reason why it is good to first burn-in and then start the proposed sampling/weighing schemes? Further comments: - since the authors propose a practical algorithm it would be nice to include a sentence about memory requirement or other non-trivial implementation details (in case there are some). - As mentioned above, I encourage the authors to expand the technical section (3.2 before the logReg example and especially 3.3) since it is very dense. - It is very nice that the authors compare against many baselines - It is nice that the authors are transparent about how the learning rates etc. are chosen. (*after rebuttal*: The rebuttal clarified most of my questions. If the paper gets accepted I encourage the authors to also clarity the points in the paper (to eases the reading process) as they promise in the rebuttal, especially i) the presentation above the logistic regression example in Section 3.2 (maybe a pointer that the assumptions will be discussed below). At that point they are quite ad-hoc and it is hard (and not beneficial for the flow of thoughts) to trace back later ii) closeness to threshold: recap shortly what \bar{p} was (it took me a while to find it again) iii) the meaning of the burn-in phase. iv) A sentence about implementation which points to the Appendix. In good faith that this will be done I increased the score.)

Reviewer 3



This paper mainly introduces two methods in the SGD framework to emphasize high variance samples in training. One is to measures the variance prediction probabilities and the other is to estimate the closeness between the prediction probabilities and the decision threshold. I think the methods proposed by this paper is simple and clean. They also make sense intuitively. In the experiment sections, the authors did a solid analysis (in 10 datasets) to test the performance of the proposed methods. These give clear evidence that the proposed methods based on the prediction variance and the threshold closeness, despite mathematically quite simple, achieve consistently better results than reasonable baselines (SGD-SCAN, SGD-WD and SGD-WE). They also consider the situation where the training dataset is more/less noisy and the results given by the experiments matches the intuition of the proposed algorithms.